# Severity of infection with the SARS-CoV-2 B.1.1.7 lineage among hospitalized COVID-19 patients in Belgium

**Nina Van Goethem**[1,2]*, **Mathil Vandromme**[1], **Herman Van Oyen**[1], **Freek Haarhuis**[1], **Ruben Brondeel**[1], **Lucy Catteau**[1], **Emmanuel André**[3,4], **Lize Cuypers**[3], **Belgian Collaborative Group on COVID-19 Hospital surveillance**[¶], **COVID-19 Genomics Belgium consortium**[¶], **Koen Blot**[1], **Ben Serrien**[1]

1 Scientific Directorate of Epidemiology and Public Health, Sciensano, Brussels, Belgium, 2 Department of Epidemiology and Biostatistics, Institut de recherche expérimental et clinique, Faculty of Public Health, Université catholique de Louvain, Woluwe-Saint-Lambert, Belgium, 3 University Hospitals Leuven, Department of Laboratory Medicine, National Reference Centre for Respiratory Pathogens, Leuven, Belgium, 4 KU Leuven, Department of Microbiology, Immunology and Transplantation, Laboratory Clinical Bacteriology and Mycology, Rega Institute for Medical Research, Leuven, Belgium

¶ Membership of the Belgian Collaborative group on COVID-19 Hospital surveillance is listed in the Acknowledgments.
¶ Membership of the COVID-19 Genomics Belgium consortium is listed in the Acknowledgments.
* nina.vangoethem@sciensano.be

**Data Availability Statement:** The individual level datasets generated or analyzed during the current study do not fulfill the requirements for open data access. The data is too dense and comprehensive

## Abstract

### Introduction

The pathogenesis of COVID-19 depends on the interplay between host characteristics, viral characteristics and contextual factors. Here, we compare COVID-19 disease severity between hospitalized patients in Belgium infected with the SARS-CoV-2 variant B.1.1.7 and those infected with previously circulating strains.

### Methods

The study is conducted within a causal framework to study the severity of SARS-CoV-2 variants by merging surveillance registries in Belgium. Infection with SARS-CoV-2 B.1.1.7 ('exposed') was compared to infection with previously circulating strains ('unexposed') in terms of the manifestation of severe COVID-19, intensive care unit (ICU) admission, or in-hospital mortality. The exposed and unexposed group were matched based on the hospital and the mean ICU occupancy rate during the patient's hospital stay. Other variables identified as confounders in a Directed Acyclic Graph (DAG) were adjusted for using regression analysis. Sensitivity analyses were performed to assess the influence of selection bias, vaccination rollout, and unmeasured confounding.

### Results

We observed no difference between the exposed and unexposed group in severe COVID-19 disease or in-hospital mortality (RR = 1.15, 95% CI [0.93–1.38] and RR = 0.92, 95% CI [0.62–1.23], respectively). The estimated standardized risk to be admitted in ICU was

to preserve patient privacy. The protocol of the LINK-VACC project was approved by the medical ethics committee University Hospital Brussels – Vrije Universiteit Brussel (VUB) on 03/02/2021 (reference number 2020/523) and obtained authorization from the Information Security Committee (ISC) Social Security and Health (reference number IVC/KSZG/21/034). The data of the individual data sources (Clinical Hospital Survey, Vaccinnet+, COVID-19 TestResult Database, and StatBel) within the LINK-VACC project are kept in the pseudonymized environment of healthdata.be and a link between the individual data in each of them takes place thanks to the use of a pseudonymized national reference number managed by healthdata.be under a "project mandate". A "project mandate" consists of a group of individuals, a group of variables and a time period. Access rights to the pseudonymized data in the healthdata.be data warehouse are granted ad nominatum for the scientists involved in the surveillance activities at Sciensano. External investigators with a request for selected data should send a proposal to covacsurv@sciensano. be. Depending on the type of desired data (anonymous or pseudonymized) the provision of data will have to be assessed by the Belgian Information Security Committee Social Security & Health based on legal and ethical regulations, and is outlined in a data transfer agreement with the data owner (Sciensano).

**Funding:** The author(s) received no specific funding for this work.

**Competing interests:** The authors have declared that no competing interests exist.

significantly higher (RR = 1.36, 95% CI [1.03–1.68]) when infected with the B.1.1.7 variant. An age-stratified analysis showed that among the younger age group ($\leq$65 years), the SARS-CoV-2 variant B.1.1.7 was significantly associated with both severe COVID-19 progression and ICU admission.

## Conclusion

This matched observational cohort study did not find an overall increased risk of severe COVID-19 or death associated with B.1.1.7 infection among patients already hospitalized. There was a significant increased risk to be transferred to ICU when infected with the B.1.1.7 variant, especially among the younger age group. However, potential selection biases advocate for more systematic sequencing of samples from hospitalized COVID-19 patients.

## Introduction

Coronavirus disease 19 (COVID-19) resulting from infection with the severe acute respiratory syndrome coronavirus 2 (SARS-CoV-2) has caused a major worldwide pandemic and public health crisis since its spread to the human population by the end of 2019 in Wuhan, China [1, 2]. The clinical spectrum of COVID-19 disease ranges from asymptomatic or mild respiratory tract illness to severe pneumonia and acute respiratory distress syndrome (ARDS) [3–6]. Also among hospitalized COVID-19 patients, outcomes significantly differ between patients, across settings, and over the course of the epidemic [7–10]. The pathogenesis of COVID-19 is complex and multifactorial. The clinical impact depends on the interplay between host characteristics [11–13], including age, certain comorbidities and genetic predisposition [14, 15], vaccination [16–18] and therapeutics [19–21], healthcare organizational aspects [22], and viral characteristics [23–25].

Next-generation sequencing (NGS) technologies make routine pathogen whole-genome sequencing (WGS) accessible at the population level in high throughput within short time frames [26–28], and has been extensively applied during the COVID-19 pandemic [29]. SARS-CoV-2, as other RNA viruses, evolves continuously. Most emerging variants will not provide a selective advantage to the virus, however some can be of concern in terms of contagiousness, vaccine escape or pathogenicity. The first Variant-Of-Concern (VOC) emerged in September 2020 in the United Kingdom (UK) [30, 31] (labeled as alpha-variant, 20I/501Y.V1 or B.1.1.7) and has several mutations including one in the Receptor Binding Domain (RBD) of the Spike (S) protein at position 501 (N501Y). Since December 2020, numerous other countries also reported cases of the B.1.1.7 lineage. By half of January it became the dominant circulating variant in the European Union (EU) [32], likely related to its increased transmissibility [33–35]. Indeed, evidence from epidemiological studies suggests that the B.1.1.7 variant is 43–90% more transmissible than pre-existing variants [33] and that the B.1.1.7 variant increases the effective reproduction number by a factor 1.5–2.0 [36]. Belgium experienced multiple travel-related introductions of the B.1.1.7 variant, particularly in patients diagnosed around the Christmas holidays [37]. After a constant rise in proportion starting from January 2021, the B.1.1.7 lineage became the dominant lineage and represented more than half of all analyzed samples by the end of February 2021 [38, 39], which was followed by a third epidemic wave. Its enhanced transmissibility may be reflected by an increase in viral load [40, 41] and a high viremia may have a role in disease pathogenesis as is the case for other respiratory viral infections

[42–45]. Indeed, the concern had been raised that B.1.1.7 also causes more severe disease compared to previously circulating strains by the UK New and Emerging Respiratory Virus Threats (NERVTAG) group in January 2021 [46]. However, studies investigating the association between B.1.1.7 and disease severity were often inconclusive or had conflicting results. The updated NERVTAG report [47] underlined the potential limitations of used datasets in terms of representativeness, power, potential biases in case ascertainment, selection, unmeasured confounders, and secular trends.

Continuous genomic surveillance enables the detection of emerging genetic variants. Information on the estimated risk of a new variant causing more or less severe illness can assist clinicians to make prognoses. Moreover, it is important information for policy makers to issue guidelines, control transmission, and prepare the healthcare system by safeguarding healthcare capacity. Van Goethem *et al* [48] presented a conceptual framework to study the effect of SARS-CoV-2 variants on the severity of COVID-19 disease in hospitalized patients and described how the causal effect of variants may be estimated from data that is gathered in Belgium in the context of routine COVID-19 surveillance systems. In this study, we apply this framework to examine the effect of the B.1.1.7 lineage on disease severity among hospitalized COVID-19 patients.

## Materials and methods

The study is conducted within a causal conceptual framework to assess the effect of SARS-CoV-2 variants on COVID-19 disease severity among hospitalized patients [48]. The study protocol has been registered on Open Science Framework (OSF) prior to data analysis (registration date July, 16th 2021, DOI: 10.17605/OSF.IO/ZG3DJ). This manuscript is reported according to the STROBE guidelines [49].

### Data sources

Sciensano, the Belgian national institute for health, has initiated the LINK-VACC project, which allows linking of selected variables from existing COVID-19 registries through the national registry number, including data on hospitalized COVID-19 patients from the Clinical Hospital Survey (CHS) [50], laboratory test results (polymerase chain reaction (PCR) tests, rapid antigen tests, and sequencing information) from the COVID-19 TestResult Database [51], administered COVID-19 vaccines from the national vaccine registry (Vaccinnet+), and socio-economic information from the Belgian Statistical Office (StatBel). Data on the hospital bed occupancy was derived from the Surge Capacity Survey (SCS) [50]. Details on the different data sources and its use within the proposed conceptual framework have been described elsewhere [48].

### Study population

The study population consists of hospitalized COVID-19 patients who were admitted in a Belgian hospital from 31st August 2020 onwards and for whom an admission form was reported in the CHS up to August 9th 2021. The analysis was restricted to those with a laboratory-confirmed COVID-19 infection (RT-PCR and/or rapid antigen test). Patients that were transferred, readmitted, or hospitalized in a hospital without an intensive care unit (ICU) were excluded. Patients admitted during the first wave (i.e., admitted before August 31st 2020) were excluded and the study period corresponds to the second and third wave of the COVID-19 epidemic in Belgium. Indeed, protocols, treatment regimens and professional experience of healthcare personnel have substantially changed between the first and second wave, and are considered to be more comparable between the second and subsequent waves.

## Exposure

Infection with the SARS-CoV-2 VOC B.1.1.7 ("alpha-variant"; exposed group) was compared to infection with previously circulating SARS-CoV-2 strains (unexposed group). Exposure to B.1.1.7 was identified through WGS (i.e., confirmed B.1.1.7 samples) obtained from both baseline and active genomic surveillance, and the subsequent registration of the Pangolin lineage [52] B.1.1.7 in the COVID-19 TestResult Database. As such, the exposed group consisted of hospitalized COVID-19 patients with an admission form registered in the CHS and identified as being infected with the B.1.1.7 variant through linkage with the COVID-19 TestResult Database based on the national registry number. Patients of whom the sample was compatible with a known VOC, as obtained through presumptive genotyping without WGS confirmation, were not considered for the current analysis. To assure that the hospital admission was related to the detected infection with the B.1.1.7 variant, patients with a sample collected more than 14 days before hospital admission or collected after hospital discharge were excluded. The unexposed group consists of COVID-19 patients with an admission form registered in the CHS and diagnosed and admitted to the hospital before December 1st 2020, therefore considered to be infected with previously circulating SARS-CoV-2 strains. According to GISAID's EpiCoV database, the first identified B.1.1.7 variant in Belgium dates back to November 30th 2020 (sample date). Therefore, it is highly unlikely that patients hospitalized before December 1st 2020 were infected with the B.1.1.7 variant.

## Study design

The study is an observational multi-center matched cohort study where COVID-19 hospitalized patients are followed-up from hospital admission until death or hospital discharge and for whom information was obtained by merging different national surveillance systems based on the national registry number. The unexposed group was matched to the exposed group based on the hospital and the mean ICU occupancy rate during the hospital stay of the patient in order to assure similar levels of care between both exposure groups, as an oversaturated ICU was previously shown to impact in-hospital mortality [22].

## Outcome

The primary outcome among the hospitalized study population is the development of severe COVID-19 defined as the presence of either ICU admission, ARDS, or in-hospital death. ICU-admission and in-hospital mortality have also been analyzed as two secondary outcomes.

## Confounding

The conceptual framework as described by Van Goethem *et al* [48] used Directed Acyclic Graphs (DAGs) to represent the assumptions and limitations for estimating the causal effect of SARS-CoV-2 variants on disease severity by means of observational data gathered from routine COVID-19 surveillance systems in Belgium. Several potential confounders of the variant-severity relationship have been identified within the conceptual framework [48] and should be adjusted for to estimate a causal effect. The variables identified as direct confounders in the DAG were adjusted for using regression analysis whereas the indirect confounders (hospital and ICU occupancy rate) were adjusted for using matching.

## Statistical analyses

Matching was done using the *MatchIt* package [53]. Patients in the exposed group were matched to patients from the unexposed group on the hospital in which they were admitted

and on the average ICU bed occupancy rate (defined as the number of COVID-19 ICU patients in the hospital divided by the hospital's number of recognized ICU beds) during their hospital stay. Exact matching with a rounding to 5% of the ICU occupancy rate was used, as this resulted in the least loss of subjects while maintaining comparable levels of care between exposed and unexposed matches. Demographic and clinical information of the matched study population was presented per exposure status.

Twenty-fold multiple imputation of missing values was performed using the *mice* package [54] for all covariates (see Table 1) used in the multivariable model (see further) and for all outcomes. Binary, categorical and numerical variables were imputed with logistic regression, multinomial regression and predictive mean matching, respectively. The primary outcome, disease severity, is an indicator based on three original variables and was passively imputed and not used as predictor for missing values on its components. The imputation was performed using thirty iterations to achieve good convergence of the MCMC and the visit sequence was set from low to high proportion of missing data.

Regression standardization [55] was done using a weighted logistic model (using matching weights) with the following covariates: SARS-CoV-2 variant, age, gender, ethnicity, comorbidities (cardiovascular disease, hypertension, solid cancer, hematological cancer, chronic lung disease, chronic kidney disease, chronic liver disease, chronic neurological disease, cognitive disorder, diabetes, obesity, immunocompromised), place of infection (community, hospital, nursing home), socio-economic variables (education level at the individual level, and population density and median taxable income in the postcode of residence), vaccination status at diagnosis (no vaccination, partially vaccinated, fully vaccinated), and two-way interactions of these variables with the SARS-CoV-2 variant. Numeric variables were entered in the model with linear and quadratic terms. The causal effect was estimated with a relative risk (RR) and a risk difference (RD). Block bootstrapping [56] of matched pairs (B = 1000 replications) was done on each imputed dataset [57] to estimate the variance on each parameter of interest. Pooled point estimates and confidence intervals were then obtained using Rubin's rules for multiple imputation [58].

A stratified analysis according to age group ($\leq 65$ and $>65$ years old) was performed and considered as an exploratory analysis as it has not been pre-specified in the protocol.

All analyses were conducted in R 4.0.1 [59].

## Sensitivity analyses

A first sensitivity analysis was performed including only WGS results obtained from baseline unbiased surveillance (i.e., without active selection of specific patient groups as explained in detail in the causal framework of Van Goethem *et al* [48]). A second sensitivity analysis was performed including only patients that had not received a first vaccination dose before their COVID-19 diagnosis. The same modeling procedure as above was conducted on these two populations. Thirdly, robustness of the results to potential unmeasured or uncontrolled confounding and selection bias was assessed using the *EValue* package and summarized using the multi-bias E-value [60, 61]. The E-value is defined as the minimum strength of association, on the risk ratio scale, that an unmeasured confounder would need to have with both the treatment and the outcome to fully explain away a specific treatment-outcome association, conditional on the measured covariates [60].

## Assessment of selection bias

Potential selection bias was assessed by comparing baseline characteristics and outcomes between patients with and without available SARS-CoV-2 variant information (confirmed, i.e.

**Table 1. Baseline characteristics per exposure status within a multi-center matched cohort study to assess the impact of SARS-CoV-2 variants on COVID-19 disease severity among hospitalized patients in Belgium.**

| | Patients infected with B.1.1.7 (n = 500) | | | Patients infected with previously circulating strains (n = 3,419) | | |
|---|---|---|---|---|---|---|
| | | % | n | | % | n |
| **Demographics** | | | | | | |
| Age (years), median (IQR) | 63 (50–76) | | 500 | 71 (55–82) | | 3417 |
| Male gender, n (%) | 276 | 55.2 | 500 | 1847 | 54.0 | 3419 |
| Nursing home resident, n (%) | 23 | 4.7 | 491 | 318 | 9.6 | 3297 |
| Ethnicity, n (%) | | | | | | |
| European | 383 | 84.4 | 454 | 2422 | 80.5 | 3007 |
| North-African | 39 | 8.6 | 454 | 368 | 12.2 | 3007 |
| Sub-Saharan African | 11 | 2.4 | 454 | 98 | 3.3 | 3007 |
| Asian | 13 | 2.9 | 454 | 53 | 1.8 | 3007 |
| Hispanic | 7 | 1.5 | 454 | 44 | 1.5 | 3007 |
| **Comorbidities** | | | | | | |
| Cardiovascular Disease, n (%) | 150 | 30.1 | 498 | 1130 | 33.1 | 3415 |
| History of Arterial Hypertension, n (%) | 163 | 32.7 | 498 | 1371 | 40.1 | 3415 |
| Diabetes mellitus, n (%) | 103 | 20.7 | 498 | 849 | 24.9 | 3415 |
| Obesity, n (%) | 81 | 16.3 | 498 | 442 | 12.9 | 3415 |
| Chronic Pulmonary Disease, n (%) | 83 | 16.7 | 498 | 496 | 14.5 | 3415 |
| Chronic Neurological Disease, n (%) | 27 | 5.4 | 498 | 255 | 7.5 | 3415 |
| Chronic Cognitive Deficit, n (%) | 27 | 5.4 | 498 | 356 | 10.4 | 3415 |
| Chronic Renal Disease, n (%) | 65 | 13.1 | 498 | 457 | 13.4 | 3415 |
| Chronic Liver Disease, n (%) | 10 | 2.0 | 498 | 82 | 2.4 | 3415 |
| Solid Cancer, n (%) | 46 | 9.2 | 498 | 373 | 10.9 | 3415 |
| Haematological Cancer, n (%) | 12 | 2.4 | 498 | 68 | 2.0 | 3415 |
| Chronic Immunosuppression, n (%) | 22 | 4.4 | 498 | 69 | 2.0 | 3415 |
| **Socio-economic status** | | | | | | |
| Education level[a], n (%) | | | | | | |
| Lower | 78 | 23.2 | 336 | 647 | 26.9 | 2408 |
| Lower secondary | 85 | 25.3 | 336 | 719 | 29.9 | 2408 |
| Higher secondary | 105 | 31.3 | 336 | 587 | 24.4 | 2408 |
| Post-secondary higher education | 68 | 20.2 | 336 | 455 | 18.9 | 2408 |
| Population density[b], median (IQR) | 830 (350–2600) | | 488 | 1500 (590–2600) | | 3237 |
| Median taxable income per capita[c], median (IQR) | 27000 (24000–28000) | | 488 | 26000 (23000–28000) | | 3237 |
| **Exposure** | | | | | | |
| Place of infection, n (%) | | | | | | |
| Community-acquired | 430 | 87.9 | 489 | 2609 | 79.5 | 3281 |
| Hospital-acquired[d] | 41 | 8.4 | 489 | 373 | 11.4 | 3281 |
| Nursing home-acquired | 18 | 3.7 | 489 | 299 | 9.1 | 3281 |
| **Vaccination status** | | | | | | |
| Vaccination category[e], n (%) | | | | | | |
| Pre-vaccination | 433 | 86.6 | 500 | 3419 | 100.0 | 3419 |
| Partial vaccination | 38 | 7.6 | 500 | 0 | 0.0 | 3419 |
| Post-vaccination | 29 | 5.8 | 500 | 0 | 0.0 | 3419 |
| **Disease characteristics** | | | | | | |
| Fever at admission, n (%) | 251 | 50.2 | 500 | 1575 | 46.1 | 3414 |
| Viral syndrome at admission, n (%) | 222 | 44.4 | 500 | 1173 | 34.4 | 3414 |
| Lower respiratory symptoms at admission, n (%) | 357 | 71.4 | 500 | 1996 | 58.5 | 3414 |

(*Continued*)

**Table 1.** (Continued)

| | Patients infected with B.1.1.7 (n = 500) | | | Patients infected with previously circulating strains (n = 3,419) | | |
|---|---|---|---|---|---|---|
| | | % | n | | % | n |
| Upper respiratory symptoms at admission, n (%) | 56 | 11.2 | 500 | 291 | 8.5 | 3414 |
| Gastrointestinal symptoms at admission, n (%) | 141 | 28.2 | 500 | 772 | 22.6 | 3414 |
| Anosmia at admission, n (%) | 35 | 7.0 | 500 | 223 | 6.5 | 3414 |
| CRP (mg/l) on admission, median (IQR) | 65 (27–120) | | 474 | 52 (17–110) | | 3141 |
| Lymphocytes (/mm$^3$) on admission, median (IQR) | 750 (270–1200) | | 451 | 940 (540–1400) | | 2928 |
| LDH (U/l) on admission, median (IQR) | 340 (250–470) | | 413 | 320 (250–450) | | 2771 |
| PaO$_2$ (mmHg) on admission, median (IQR) | 65 (58–74) | | 350 | 65 (58–74) | | 1921 |
| **Outcomes** | | | | | | |
| Severe[f] COVID-19, n (%) | 149 | 30.2 | 493 | 938 | 27.7 | 3389 |
| ICU transfer, n (%) | 115 | 23.1 | 498 | 520 | 15.2 | 3415 |
| In-hospital mortality, n (%) | 61 | 12.3 | 495 | 547 | 16.1 | 3407 |
| Invasive ventilation, n (%) | 31 | 6.2 | 500 | 221 | 6.5 | 3417 |
| ECLS, n (%) | 4 | 0.8 | 500 | 24 | 0.7 | 3418 |
| Hospital length of stay (days), median (IQR) | 8 (5–17) | | 500 | 9 (5–19) | | 3419 |

CRP: C-reactive protein; ECLS: Extracorporeal life support; ICU: intensive care unit; IQR: inter-quartile range; LDH: lactate dehydrogenase; PaO2: partial blood oxygen pressure.

[a] Highest degree obtained. ED1: lower; ED2: lower secondary; ED3: higher secondary; ED5: higher.

[b] Population density at the postal code level of the residence of the patient.

[c] Median net taxable income per capita at the postal code level of the residence of the patient.

[d] Symptom onset or diagnosis more than 8 days after hospital admission.

[e] Pre-vaccination: diagnosed when no dose received or before 14 days after the first dose (for Pfizer/BioNTech, AstraZeneca and Moderna vaccine); Partial vaccination: diagnosed 14 days after the first dose (for Pfizer/BioNTech, AstraZeneca and Moderna vaccine) but before 14 days after the full dose (2 doses for Pfizer/BioNTech, AstraZeneca and Moderna vaccine and 1 dose for Johnson & Johnson vaccine); Post-vaccination: diagnosed ≥14 days after the full dose (2 doses for Pfizer/BioNTech, AstraZeneca and Moderna vaccine and 1 dose for Johnson & Johnson vaccine).

[f] Defined as a combination of three binary severity indicators: having been admitted to ICU or developed acute respiratory distress syndrome (ARDS) and/or died in the hospital.

via WGS) obtained through baseline surveillance. The comparison was conducted among patients diagnosed from March 1st 2021 onwards, as the majority of sequencing results were available in the COVID-19 TestResult database starting from this date and this cut-off subsequently leads to a comparable distribution of patients over time in both groups. Assuming that the majority of hospitalized patients had a B.1.1.7 variant during this time period [38], this comparison helps to assess whether there is a difference in profiles of patients of whom samples were or were not sequenced (e.g., due to a higher viral load or preferential sequencing).

### Ethics and data protection authorization

Ethical approval was granted for the gathering of data from hospitalized patients by the Committee for Medical Ethics from the Ghent University Hospital (reference number BC-07507) and authorization for possible individual data linkage using the national register number from the Information Security Committee (ISC) Social Security and Health (reference number IVC/KSZG/20/384). Linkage of hospitalized patient data to vaccination, testing, sequencing and socioeconomic data within the LINK-VACC project was approved by the medical ethics committee UZ Brussels–VUB on 03/02/2021 (reference number 2020/523) and obtained authorization from the ISC Social Security and Health (reference number IVC/KSZG/21/034).

Inform consent was waved based on art 6 and 9 of the GDPR. The collection is allowed based on general interest (art. 6 GDPR) and regarding article 9 § 2of the GDPR: processing is necessary for reasons of public interest in the area of public health, such as protecting against serious cross-border threats to health or ensuring high standards of quality and safety of health care and of medicinal products or medical devices, on the basis of Union or Member State law which provides for suitable and specific measures to safeguard the rights and freedoms of the data subject, in particular professional secrecy.

## Results

### Basic descriptive characteristics of the matched study population

As recorded on August 9th 2021, the CHS database contained a total of 73,370 case records of COVID-19 patients, of which admission forms were received for 67,948 patients (Fig 1). After exclusion of patients not meeting inclusion criteria, a total of 35,558 hospitalized COVID-19

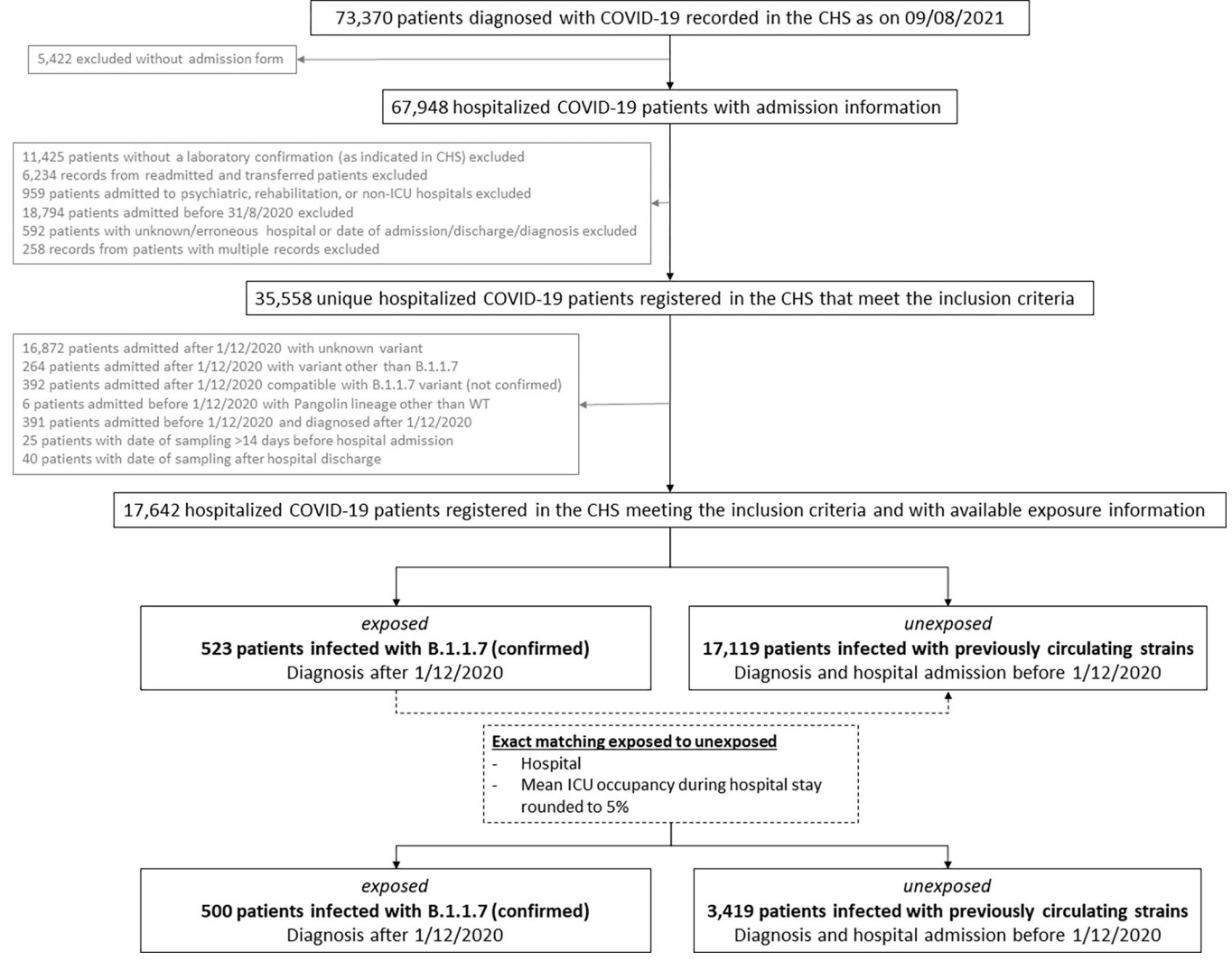

**Fig 1. Flow chart for the selection of patients within a multi-center matched cohort study to assess the impact of SARS-CoV-2 variants on COVID-19 disease severity among hospitalized patients in Belgium.**

patients were recorded as admitted after August 31st 2020 and 17,642 (49.6%) of them had available exposure information. These were either identified as having a confirmed B.1.1.7 infection (n = 523; exposed) upon linkage with the COVID-19 TestResult Database, or classified as unexposed (n = 17,119), meaning that they were diagnosed and admitted before December 1st 2020 and consequently considered as being infected with previously circulating SARS-CoV-2 strains. From the 523 patients with a confirmed B.1.1.7 infection (exposed), 500 could be matched to 3,419 patients infected with previously circulating strains (unexposed) based on the hospital and the mean ICU occupancy rate rounded to 5% and a total of 3,919 cases were thus included in the descriptive analysis.

Median age was 70 years (IQR 54–82), 54% (2,123/3,919) were male, and 9% (341/3,788) were nursing home residents. A total of 2,324 (59%) patients were admitted to a general hospital, 1,016 (26%) to a general hospital with university character, and 579 (15%) to a university hospital. The median hospital length of stay was 9 days (IQR 5–19). The median ICU bed occupancy rate (i.e., the number of recognized ICU beds occupied by COVID-19 patients) averaged over the patient's hospital stay was 35% (IQR 23–47%). The overall occurrence of severe COVID-19, ICU-admission, and in-hospital mortality, were 28% (1,087/3,882), 16% (635/3,913), and 16% (608/3,902), respectively.

Table 1 shows patient characteristics by exposure status. Patients with a confirmed B.1.1.7 infection (exposed) were diagnosed between December 23rd 2020 and July 23rd 2021, whereas patients infected with previously circulating strains (unexposed) were diagnosed before November 30th 2020. Given these different time periods, patients infected with previously circulating strains were unvaccinated, whereas 5.8% (29/500) patients with a confirmed B.1.1.7 infection were fully vaccinated at the time of diagnosis. Patients with a confirmed B.1.1.7 infection were younger by eight years and less frequently had arterial hypertension, diabetes mellitus, and chronic cognitive deficit than patients infected with previously circulating strains. On the other hand, patients with a confirmed B.1.1.7 infection were more frequently immuno-compromised and obese. Also, there were fewer nursing home residents among patients with a confirmed B.1.1.7 infection, and they consequently contracted their infection more frequently within the community rather than in health care settings. Patients with a confirmed B.1.1.7 infection more frequently presented with symptoms at hospital admission. Using the matching weights, the hospitals and the mean ICU occupancy rate were perfectly balanced between both exposure groups.

## Causal inference estimates

Table 2 presents the causal effect estimates for infection with B.1.1.7 on the risk of severe COVID-19, ICU admission, and in-hospital mortality among hospitalized patients as compared to infection with previously circulating variants. The standardized risk (with respect to the model and the covariate distribution) of severe COVID-19 was 27.2% (95% CI [24.6–29.7]) in the hospitalized patients when infected with previously circulating variants and 31.4% (95% CI [26.0–36.8]) in the hospitalized patients when infected with the B.1.1.7 variant. The difference between both exposure groups was not statistically significant at the 5% level (RD: 4.3%, 95% CI [-1.7–10.2]; RR = 1.15, 95% CI [0.93–1.38]). The estimated standardized risk to be admitted in ICU was a significant 5.7% higher in the patients when infected with the B.1.1.7 variant (95% CI [1.0–10.4]), whereas the estimated standardized risk of in-hospital mortality was a non-significant 1.2% lower (95% CI [-6.2–3.8]).

The age-stratified analysis (Table 3) revealed that among the younger age group (≤65 years) both the risk of severe COVID-19 and ICU admission was significantly higher in the patients when infected with the B.1.1.7 variant as compared to when infected with previously

Table 2. Risk per exposure group (in %), Relative Risk (RR) and Risk Difference (RD, in %) estimates and 95% Confidence Interval (CI) for main and secondary outcomes within a multi-center matched cohort study to assess the impact of SARS-CoV-2 variants on COVID-19 disease severity among hospitalized patients in Belgium.

| Outcome | Risk [a] (in %) [95% CI] | | RR [95% CI] | RD (in %) [95% CI] |
|---|---|---|---|---|
| | PCV | B.1.1.7 [b] | | |
| Severe COVID-19 [c] | 27.2 [24.6–29.7] | 31.4 [26.0–36.8] | 1.15 [0.93–1.38] | 4.3 [-1.7–10.2] |
| ICU admission | 15.9 [13.7–18.1] | 21.6 [17.5–25.7] | 1.36 [1.03–1.68] | 5.7 [1.0–10.4] |
| In-hospital mortality | 16.2 [14.0–18.4] | 15.0 [10.2–19.7] | 0.92 [0.62–1.23] | -1.2 [-6.2–3.8] |

CI: confidence interval; ICU: intensive care unit; PCV: previously circulating variants; RD: risk difference; RR: risk ratio.

[a] Standardized risk with respect to the model and covariate distribution.

[b] Confirmed via Whole-Genome Sequencing (WGS)

[c] Presence of acute respiratory distress syndrome (ARDS), ICU admission and/or in-hospital death.

circulating strains (RR = 1.55, 95% CI [1.15–1.97] and RR = 1.69, 95% CI [1.21–2.17], respectively). There was no increased risk of severe COVID-19 or ICU admission for the elderly patients (>65 years) when infected with the B.1.1.7 variant (RR = 1.04, 95% CI [0.75–1.33] and RR = 1.13, 95% CI [0.11–2.16], respectively). There was no significant increased risk of in-hospital mortality in neither of the age groups (≤65 or >65 years).

## Sensitivity analyses

A first sensitivity analysis assessed whether only including samples sequenced within the context of baseline (i.e., without active) surveillance would influence the results. S1 Fig shows a flowchart for selection of patients for whom WGS was performed in the context of baseline surveillance. From the 264 patients with a confirmed B.1.1.7 infection identified through baseline surveillance, 253 could be matched to 2,126 patients infected with previously circulating variants. The causal effect estimates within this subgroup, as presented in S1 Table, are similar compared to the main analysis results.

Table 3. Risk per exposure group (in %), Relative Risk (RR) and Risk Difference (RD, in %) estimates and 95% Confidence Interval (CI) for main and secondary outcomes, stratified per age group, within a multi-center matched cohort study to assess the impact of SARS-CoV-2 variants on COVID-19 disease severity among hospitalized patients in Belgium.

| Outcome | Risk [a] (in %) [95% CI] | | RR [95% CI] | RD (in %) [95% CI] |
|---|---|---|---|---|
| | PCV | B.1.1.7 [b] | | |
| **Age ≤ 65 years** | | | | |
| Severe COVID-19 [c] | 16.9 [14.4–19.5] | 26.4 [20.4–32.3] | 1.55 [1.15–1.97] | 9.5 [3.2–15.7] |
| ICU-admission | 14.7 [12.2–17.2] | 24.8 [18.9–30.7] | 1.69 [1.21–2.17] | 10.1 [3.9–16.3] |
| In-hospital mortality | 3.9 [2.4–5.4] | 7.3 [1.2–13.4] | 1.85 [0.16–3.55] | 3.4 [-2.9–9.6] |
| **Age > 65 years** | | | | |
| Severe COVID-19 [c] | 34.3 [30.8–37.9] | 35.8 [27.1–44.4] | 1.04 [0.75–1.33] | 1.4 [-8.3–11.1] |
| ICU-admission | 16.8 [13.9–19.8] | 19.1 [11.8–26.4] | 1.13 [0.11–2.16] | 2.3 [-5.6–10.2] |
| In-hospital mortality | 24.5 [21.3–27.8] | 24.0 [15.2–32.9] | 0.98 [0.60–1.36] | -0.5 [-9.8–8.8] |

CI: confidence interval; ICU: intensive care unit; PCV: previously circulating variants; RD: risk difference; RR: risk ratio.

[a] Standardized risk with respect to the model and covariate distribution.

[b] Confirmed via Whole-Genome Sequencing (WGS)

[c] Presence of acute respiratory distress syndrome (ARDS), ICU admission and/or in-hospital death.

The second sensitivity analysis excluded patients that had received at least one vaccination dose before their COVID-19 diagnosis, in order to account for the impact of the vaccination rollout between the exposed and unexposed group. S2 Fig shows a flow chart for selection of patients that did not receive a vaccination dose before their COVID-19 diagnosis. From the 419 patients with a confirmed B.1.1.7 infection and no vaccination dose received before diagnosis, 405 could be matched to 2,881 patients infected with previously circulating variants. The causal effect estimates within this subgroup, as presented in S2 Table, are similar compared to the main analysis results.

The E-value and multi-bias E-value were calculated to assess the influence of selection bias (e.g., based on the viral load) and/or unmeasured confounding (e.g., genetic profile of the patient) on the observed RR for each of the outcomes (S3 Table). The observed significant RR of 1.36 for ICU admission could be explained by an unmeasured confounder that was associated with both the exposure (SARS-CoV-2 variant) and ICU admission by a RR of 2.06-fold each, above and beyond the measured confounders, but weaker confounding could not do so; the confidence interval could be moved to include the null by an unmeasured confounder that was associated with both the exposure and ICU admission by a RR of 1.21-fold each, above and beyond the measured confounders, but weaker confounding could not do so. The same applies to selection on a variable with associations to both exposure and ICU transfer of at least 2.06 (1.21 for the 95% CI). A multi-bias E-value of 1.60 was obtained when considering both unmeasured confounding and selection bias simultaneously. This means that an unmeasured confounder with an association on the RR-scale of at least 1.60 to both exposure and outcome and selection on a variable with an association on the RR-scale of at least 1.60 to both exposure and outcome could explain the observed effect (above and beyond the variables that were controlled for in the model).

## Selection bias

Selection bias was assessed by comparing the differences between patients of whom the SARS-CoV-2 positive sample was or was not selected for WGS analysis. Patients of whom the sample was compatible with a known VOC, as obtained through presumptive genotyping without WGS confirmation, were excluded. From the 9,599 patients with an available admission form registered in the CHS, meeting the inclusion criteria, and admitted in the hospital after March 1$^{st}$ 2021, 672 (7%) had a sample with a confirmed Pangolin lineage. About half of those sequencing results (53%; 357/672) were obtained through baseline surveillance. S4 Table compares patients with variant information (obtained through baseline WGS surveillance) to patients without SARS-CoV-2 variant information. Patients for whom baseline WGS surveillance was performed were more frequently males, nursing home residents, immunocompromised, fully vaccinated, admitted to a university hospital, and contracted their infection more frequently within the hospital. Moreover, these patients were more frequently transferred to ICU as compared to patients without available sequence information. When stratifying per hospital type, patients in general hospitals with viral sequence data were more frequently admitted into ICU as compared to patients without viral sequence data (20.9%; 95% CI [15.9%– 26.8%] and 13.7%, 95% CI [12.8%– 14.5%], respectively), whereas this difference was not observed among patients admitted to general hospitals with university characteristics or university hospitals.

## Discussion

This study aimed to assess the effect of the SARS-CoV-2 VOC B.1.1.7 (also labeled as alpha-variant) on disease severity among hospitalized COVID-19 patients within an existing causal

framework [48] using linked data from routine COVID-19 surveillance systems in Belgium. We observed no significant difference in severe COVID-19 disease or in-hospital mortality by SARS-CoV-2 lineage (B.1.1.7 versus non-sequenced previously circulating variants) in an adjusted analysis (RR = 1.15, 95% CI [0.93–1.38] and RR = 0.92, 95% CI [0.62–1.23], respectively). This is in line with the findings from Frampton *et al* [40] where no association was found between B.1.1.7 infection and severe disease or death within a hospitalized cohort. On the other hand, community-based studies revealed an increased risk of overall mortality associated with B.1.1.7 in people testing positive for COVID-19 [62–65]. These findings may suggest that the effect of B.1.1.7 is different in a hospitalized cohort than in the general population and does not exclude an increased risk of hospital admission with the B.1.1.7 lineage [64]. Indeed, a Danish [66] and two UK [67, 68] studies suggested that infection with lineage B.1.1.7 was associated with an increased risk of hospitalization compared with that of other circulating strains or the wild-type variant. As such, it is possible that the B.1.1.7 variant has an increased risk of hospitalization, but that there is no additive risk of mortality once hospitalized [40, 64, 69]. However, restricting the analysis to hospitalized patients may induce collider bias [70, 71]. Among hospitalized patients, the relationships between any variables that relate to hospitalization will be distorted compared to the relationships that exist among the general population [70]. As such, the identified associations within the hospitalized population may not reflect the patterns in the general population (i.e., lack of external validity) [71].

The estimated standardized risk to be admitted in ICU was significantly higher (RR = 1.36, 95% CI [1.03–1.68]) in the patients when infected with the B.1.1.7 variant. This is in line with the findings from a community-based study by Patone *et al* who reported that people infected with lineage B.1.1.7 had double the risk of admission to ICU compared to those infected with non-B.1.1.7 SARS-CoV-2 [64]. However, we should be aware that selection bias could potentially invalidate our causal inference estimates [61]. Here, we observed that patients with variant information available differ from patients of whom the samples were not selected for WGS analysis. As such, 22% of hospitalized patients with available sequencing results were transferred to ICU, whereas this was only the case for 16% of hospitalized patients without information on the SARS-CoV-2 lineage of their infection. This could in part be explained by the fact that patients with available sequence information were more often admitted to a university hospital where the proportion of ICU transfers is higher. However, given our matched cohort design, the type of hospital is perfectly balanced between the exposed and unexposed group and should not result in confounding. Furthermore, the model also matches patients based on levels of ICU occupancy, as patients may less likely be admitted when ICU capacity is oversaturated. Still, selection bias may arise when the samples from ICU patients are preferentially selected for WGS. Indeed, if a nonrandom selection of samples for WGS based on the severity of disease or ICU admission occurs, this may partially explain why we observed a higher standardized risk for ICU admission for patients with a confirmed B.1.1.7 infection compared to patients without available sequencing results that were considered to be infected with previously circulating strains. However, a sensitivity analysis considering only sequencing results obtained through baseline (unbiased) surveillance provided similar results. Another potential source of bias is the fact that only samples with a sufficiently high viral load ($\geq 10^3$–$10^4$ RNA copies/mL) can be sequenced due to technical limitations. This could bias our conclusions, as a higher viral load can be associated with severe disease [42]. However, the viral load also depends on the stage in which the patient is sampled (time of sampling) and the underlying conditions of the patients. Here, the robustness of our obtained causal inference estimates to potential uncontrolled confounding, such as the viral load, was assessed using the E-value [60]. If both the association between viral load and exposure (i.e., SARS-CoV-2 variant) and the association between viral load and ICU transfer, is at least 2.06 on the risk ratio scale

(conditional on the other included covariates), this could completely nullify the observed causal estimate (RR = 1.36, 95% CI [1.03–1.68]) to be admitted in ICU. This relatively large E-value implies that considerable unmeasured or uncontrolled confounding would be needed to explain away our obtained effect estimate.

Our exploratory analyses revealed important differences in the risk for severe COVID-19 and ICU admission associated with the B.1.1.7 variant according to age. We did observe an increased risk of severe COVID-19 related to the B.1.1.7 variant among the younger age group (≤65 years), whereas severity seemed to be independent of the SARS-CoV-2 variant among the older age group (>65 years). This is line with an analysis based on data from seven EU countries that also suggests a higher risk for hospitalization and ICU admission in age groups <60 years for B.1.1.7, whereas this was not the case for the older age groups [72]. One hypothesis to explain these observations is that the B.1.1.7 variant causes a higher viral load [40] as compared to previously circulating variants, but that the positive correlation between viral load and disease severity is only observed in younger patients. Indeed, it has been shown that respiratory viral loads were generally correlated with inflammatory responses in younger patients, but less correlated with those in older patients [73].

Within the current study, the exposed and unexposed group are completely separated in time. As a limitation, the unexposed in the analysis did not have information (obtained through WGS) on the SARS-CoV-2 variant of their infection. They were defined as being infected with 'previously circulating strains' as they were diagnosed and admitted in the hospital before December 1st 2020, i.e., before the circulation of any VOC in Belgium. However, we cannot rule out the possibility that a patient was hospitalized in Belgium after being infected with a VOC abroad. As large-scale COVID-19 genomic surveillance was initiated when B.1.1.7 became dominant in Belgium, there were insufficient sequenced non-VOC samples from patients hospitalized after December 1st 2020 to facilitate comparisons. Given the different time periods and the non-randomized observational study design, the exposed and unexposed groups considerably differ in terms of patient characteristics and contextual factors. The profile of hospitalized patients may change over time according to the demography of the viral circulation. Indeed, the patients in the exposed group were younger (in line with Frampton *et al* [40]), which may also explain the differences in distributions of comorbidities, illness severity, and presenting symptoms at admissions between both groups. Given the different time periods, there may be an impact of the vaccination rollout in Belgium which started in early 2021 and targeted in priority the nursing home residents, healthcare workers, and people with comorbidities. However, a sensitivity analysis excluding the vaccinated patients provided similar results. Further, although there were no apparent changes in national or regional policies, there may exist differences in indications for hospitalization of COVID-19 patients between the two time periods related to the number of available beds and medical personnel. However, we believe that matching the exposure groups based on the mean ICU occupancy rate (calculated as the number of COVID-19 patients occupying the recognized ICU beds within the hospital in which the patient was admitted and averaged over the patient's hospital stay) controlled well for the risk of hospital or ICU admission related to organizational characteristics. In addition, matching on the hospital enables to account for between-hospital differences in admission criteria and levels of care. Moreover, the decision-making process to admit COVID-19 patients may also be influenced by individual patient characteristics such as age. Therefore, a major strength of the current study in general is the ability to control for an extensive list of potential confounders (i.e., patient characteristics and contextual factors that differ between the two time periods) given the detailed patient information that is collected within the CHS and the linkage to other data sources. For instance, our ability to control for the mean ICU occupancy rate is an important strength given previous observations that mortality is

affected by how many patients require intensive care in a hospital setting [22, 74]. As a limitation, we missed information on the staff to patient ratio and could not take into account the number of newly created ICU beds per hospital. Also, there may exist other time-dependent factors for which we are unable to adjust. This will in general always be an issue, as different emerging variants will become dominant consecutively over time and as there is often only a short period in which two variants co-circulate and can be directly compared. Also, in order to study the clinical impact of variants within the current framework based on linking routine COVID-19 registries, one variant may need to dominate a previous one before a sufficiently large sample size is reached. This has implications for the timeliness of the results for guiding policy making.

The limitations that we encountered with regard to potential selection biases and a sample size that depends on the linkage of existing COVID-19 registries, emphasizes the need for more systematic sequencing of samples from hospitalized COVID-19 patients. A major focus of the current genomic surveillance program is on detecting new emerging variants and flagging specific events, such as break-through cases, re-infections, and geographic dynamics by monitoring returning travelers [75]. However, a detailed analysis of the association between SARS-CoV-2 variants and disease severity requires a sufficiently large and representative sample. This could be achieved by a better alignment of the different stakeholders. For example, sequencing capacity could be efficiently redistributed by performing random or exhaustive sequencing of COVID-19 samples from hospitalized patients. This would optimize linking of multiple independent data sources in settings where this is required. Further, the indication for sequencing (i.e., baseline versus active surveillance of severe patients) should be well documented by the laboratories when reporting data in order to avoid selection biases.

## Conclusions

In this observational multi-center matched cohort study, we observed that among patients already hospitalized, no increased risk of severe COVID-19 infection or death associated with B.1.1.7 infection was found compared to previously circulating SARS-CoV-2 strains. Within an age-stratified analysis we did observe that among the ≤ 65 age group the risk for severe COVID-19 was higher among patients when infected with the B.1.1.7 variant, whereas severity was independent of the SARS-CoV-2 variant among the older age group (>65 year). Although we should take into account the risk of non-random selection of samples for WGS, we did observe an overall association with B1.1.7 infection and ICU admission. While at the moment of writing the delta-variant has completely dominated the B.1.1.7 variant [76], this analysis may still provide useful scientific information for future comparisons with new emerging variants. Performing real-time and unbiased assessments of the severity related to emerging SARS-CoV-2 variants should be foreseen in the future. Systematic screening of samples from hospitalized COVID-19 patients is needed to avoid potential biases.

## Supporting information

**S1 Fig. Flow chart for the selection of patients when only considering sequencing results obtained from baseline surveillance.** Flow chart for a sensitivity analysis within a multi-center matched cohort study to assess the impact of SARS-CoV-2 variants on COVID-19 disease severity among hospitalized patients in Belgium.
(TIF)

**S2 Fig. Flow chart for the selection of patients when excluding patients who received at least one COVID-19 vaccine dose before COVID-19 diagnosis.** Flow chart for a sensitivity

analysis within a multi-center matched cohort study to assess the impact of SARS-CoV-2 variants on COVID-19 disease severity among hospitalized patients in Belgium.
(TIF)

**S1 Table. Risk per exposure group (in %), Relative Risk (RR) and Risk Difference (RD, in %) estimates and 95% Confidence Interval (CI) for main and secondary outcomes when only considering Whole-Genome Sequencing (WGS) results obtained through baseline surveillance.** Results (overall and stratified per age group) for a sensitivity analysis within a multi-center matched cohort study to assess the impact of SARS-CoV-2 variants on COVID-19 disease severity among hospitalized patients in Belgium.
(DOCX)

**S2 Table. Risk per exposure group (in %), Relative Risk (RR) and Risk Difference (RD, in %) estimates and 95% Confidence Interval (CI) for main and secondary outcomes when excluding patients that had received a first vaccination dose before their COVID-19 diagnosis.** Results (overall and stratified per age group) for a sensitivity analysis within a multi-center matched cohort study to assess the impact of SARS-CoV-2 variants on COVID-19 disease severity among hospitalized patients in Belgium.
(DOCX)

**S3 Table. Sensitivity analysis using the E-value.** Sensitivity analysis within a multi-center matched cohort study to assess the impact of SARS-CoV-2 variants on COVID-19 disease severity among hospitalized patients in Belgium.
(DOCX)

**S4 Table. Assessment of selection bias.** Baseline characteristics between hospitalized patients in Belgium admitted after March 1st 2021 with available variant information (confirmed) obtained through baseline surveillance and without available variant information.
(DOCX)

## Acknowledgments

We are very grateful to all clinicians and hospital directions which allowed this systematic and timely data collection and reporting to Sciensano. In particular, we would like to acknowledge the Belgian Collaborative Group on COVID-19 Hospital surveillance: Amir-Samy Aouachria, Kristof Bafort, Leïla Belkhir, Nathalie Bossuyt, Steven Callens, Vincent Colombie, Sarah Cooreman, Nicolas Dauby, Paul De Munter, Pieter Depuydt, Didier Delmarcelle, Mélanie Delvallee, Rémy Demeester, Thierry Dugernier, Caroline Gheysen, Xavier Holemans, Benjamin Kerzmann, Sarah Loof, Pierre Yves Machurot, Geert Meyfroidt, Philippe Minette, Jean-Marc Minon, Saphia Mokrane, Catherine Nachtergal, Séverine Noirhomme, Denis Piérard, Camelia Rossi, Carole Schirvel, Erica Sermijn, Frank Staelens, Fabio Silvio Taccone, Frederic Thys, Filip Triest, Jens Van Praet, Eva Van Braeckel, Anke Vanhoenacker, Roeland Verstraete, Elise Willems, and Chloé Wyndham-Thomas.

We would also like to thank all sequencing laboratories for uploading SARS-CoV-2 variant information to the COVID-19 TestResult database. In particular, we would like to thank the members of the COVID-19 Genomics Belgium consortium: Emmanuel André, Piet Maes, Guy Baele, Simon Dellicour, Lize Cuypers, Marc Van Ranst, Barney Potter, Samuel Hong, François E. Dufrasne, Guillaume Bayon-Vicente, Ruddy Wattiez, Carl Vael, Lynsey Berckmans, Philippe Selhorst, Kevin K. Ariën, Arnaud Marchant, Coralie Henin, Benoit Haerlingen, Ricardo De Mendonca, Marie-Luce Delforge, Sonia Van Dooren, Bruno Hinckel, Hideo Imamura, Toon Janssen, Ben Caljon, Oriane Soetens, Denis Piérard, Thomas Demuyser, Charlotte

Michel, Olivier Vandenberg, Sigi van den Wijngaert, Giulia Zorzi, Jean Ruelle, Benoit Kabamba Mukadi, Jean-Luc Gala, Bertrand Bearzatto, Jérôme Ambroise, Philippe Van Lint, Walter Verstrepen, Reinout Naesens, Michael Peeters, Kate Bakelants, Sarah Denayer, Sofieke Klamer, Pascale Hilbert, Sylvain Brohée, Pierre-Emmanuel Léonard, Deniz Karadurmus, Jeremie Gras, Damien Féret, Barbara Lambert, Anne Vankeerberghen, Astrid Holderbeke, Hans De Beenhouwer, Lien Cattoir, Christine Lammens, Basil Britto Xavier, Marie Le Mercier, Jasmine Coppens, Veerle Matheeussen, Herman Goossens, Geert A. Martens, Koen Swaert, Frederik Van Hoecke, Dieter Desmet, Pierre Bogaerts, Jonathan Degosserie, Olivier Denis, Te-Din Huang, Dagmar Obbels, Hanne Valgaeren, Johan Frans, Annick Smismans, Paul-Emile Claus, Truus Goegebuer, Ann Lemmens, Bea Van den Poel, Sonja De Bock, Wim Laffut, Ellen Van Even, Jos Van Acker, Charlotte Verfaillie, Elke Vanlaere, Klara De Rauw, Brigitte Maes, Guy Froyen, Bert Cruys, Ellen Geerdens, Luc Waumans, Britt Van Meensel, Reinoud Cartuyvels, Severine Berden, Marijke Raymaekers, Bruno Verhasselt, Cécile Meex, Keith Durkin, Laurent Gillet, Maria Artesi, Marie-Pierre Hayette, Sébastien Bontems, Vincent Bours, Claire Gourzonès, Olivier Ek, Fabrice Bureau, Jorn Hellemans, Patrick Descheemaeker, and Marijke Reynders.

## Author Contributions

**Conceptualization:** Nina Van Goethem, Mathil Vandromme, Ruben Brondeel, Lucy Catteau, Emmanuel André, Lize Cuypers, Koen Blot, Ben Serrien.

**Data curation:** Nina Van Goethem, Mathil Vandromme, Freek Haarhuis, Ruben Brondeel, Emmanuel André, Lize Cuypers, Ben Serrien.

**Formal analysis:** Nina Van Goethem, Mathil Vandromme, Ben Serrien.

**Investigation:** Nina Van Goethem, Mathil Vandromme, Ben Serrien.

**Methodology:** Nina Van Goethem, Mathil Vandromme, Freek Haarhuis, Lucy Catteau, Ben Serrien.

**Project administration:** Lucy Catteau, Lize Cuypers.

**Supervision:** Herman Van Oyen, Ruben Brondeel, Lucy Catteau, Lize Cuypers, Koen Blot, Ben Serrien.

**Validation:** Freek Haarhuis, Ben Serrien.

**Writing – original draft:** Nina Van Goethem, Ben Serrien.

**Writing – review & editing:** Nina Van Goethem, Mathil Vandromme, Herman Van Oyen, Freek Haarhuis, Ruben Brondeel, Lucy Catteau, Emmanuel André, Lize Cuypers, Koen Blot, Ben Serrien.

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
