## [Decision Letter · Decision Letter 0]

3 Apr 2022

PONE-D-21-40413Severity of infection with the SARS-CoV-2 B.1.1.7 lineage among hospitalized COVID-19 patients in BelgiumPLOS ONE

Dear Dr. Van Goethem,

Thank you for submitting your manuscript to PLOS ONE. After careful consideration, we feel that it has merit but does not fully meet PLOS ONE’s publication criteria as it currently stands. Therefore, we invite you to submit a revised version of the manuscript that addresses the points raised during the review process.

We look forward to receiving your revised manuscript.

Kind regards,

Valérie Pittet, PhD

Academic Editor

PLOS ONE

Journal Requirements:

Reviewers' comments:

Reviewer's Responses to Questions

**Comments to the Author**

1. Is the manuscript technically sound, and do the data support the conclusions?

Reviewer #1: Yes

Reviewer #2: Yes

Reviewer #3: Yes

2. Has the statistical analysis been performed appropriately and rigorously? 

Reviewer #1: Yes

Reviewer #2: Yes

Reviewer #3: Yes

3. Have the authors made all data underlying the findings in their manuscript fully available?

Reviewer #1: No

Reviewer #2: Yes

Reviewer #3: Yes

4. Is the manuscript presented in an intelligible fashion and written in standard English?

Reviewer #1: Yes

Reviewer #2: Yes

Reviewer #3: Yes

5. Review Comments to the Author

Reviewer #1: Dear Editor,

Thank you for the opportunity to revise the manuscript PONE-D-21-40413 “Severity of infection with the SARS-CoV-2 B.1.1.7 lineage among hospitalized COVID-19 patients in Belgium.”

This is a very well written manuscript reporting results of a nationwide study on one of the more urgent topic in public health. The main value of the study is to establish a method that can be timely applied to assess the upcoming variant severity compared to previous circulating ones in hospitalised patients.

In the current version of the manuscript, aims, methods, and results were clearly reported. Discussion and conclusions were based on presented results. Moreover, the major limitations and potential bias were deeply assessed and discussed, including the intrinsic limitations of hospital-based studies.

I only have some minor comments, mainly related to clarifications on contextual factors.

1. The main comment is about indications to hospitalization in Belgium during the study period for COVID19 patients. As the authors pointed out, including only hospitalised patients is a limitation especially for external validity of study results. I think that a comment on potential differences in indications for hospitalization of COVID19 patients between the two study periods and between younger and elderly patients could increase the understanding of the internal validity of study results.

2. Using the ICU bed occupancy rate as matching variable is a good strategy to reduce the impact of healthcare related factors on differences in patients ‘outcomes. Notwithstanding, the absolute number of ICU beds even in the same hospital varied greatly across the pandemic periods, with an increase during each wave and generally inversely correlated with the quality of healthcare assistance provided (i.e. for healthcare services overload). Can you make a comment on the potential impact of this phenomenon on study results considering that it goes in the direction of higher risk for ICU admission among cases with B.1.1.7 variant, especially young?

3. Despite very interesting, the chapter reporting the E-value and multi-bias E-value results is not straightforward. Could you rewrite it or expand the related paragraph in the discussion section explaining how these results relate with the robustness of the study results?

I would like to conclude by congratulating the study authors for the impressive work done.

Best regards

Reviewer #2: It's an excellent paper that makes use of large-scale surveillance data to determine the severity of variants of concern for omicron sublineage B.1.1.7 in comparison to previously circulating strains. The findings are extremely informative, and the data was rigorously analysed. It's interesting that the authors made a concerted effort to provide "the most valid estimates" possible by adjusting for confounding variables using either matching or DAG, as used in this work. This article should be published to provide additional evidence regarding the severity of B.1.1.7 infection in a hospital-based setting.

Reviewer #3: Comments on PLOS Manuscript PONE-D-21-40413

Overall the manuscript is very well written. However, there is need for some corrections and edits.

RESULTS

277-279 - (Include Confidence Intervals (CI) in Table 1, where relevant)

325-326 - There was no significant increased risk of in-hospital in neither of the age groups (<65 or > 65) - (review language)

382-384 – When stratifying……,patients in general hospitals with B.1.1.7 infection were more frequently admitted in ICU (20.9-13.7%). - (include CI)

DISCUSSION

403 - However, restricting the analysis to hospitalized patients may include collider bias. - (include reference)

441 - ……..whereas severity seemed to be independent on the SARS- CoV-2 variant among the elderly- (review language)

481-484 - A major focus of the current genomic surveillance program is on detecting new emerging variants and flagging…………and geographic dynamics by monitoring returning travelers. - (Include reference)

CONCLUSIONS

497-498 - …….whereas severity was independent from the SARS- CoV-2 among 65-plus patients. (review language - needs clarifying for consistency- see 441-443)

500-501 - While at the moment of writing the delta-variant has completely dominated the B 1.1.7 variant, (Include ref.) this analysis……

502-503 - Real-time and unbiased analyses of emerging SARS-CoV-2 variants and their association with disease severity should be foreseen in the future. - (review language).

6. PLOS authors have the option to publish the peer review history of their article (what does this mean?). If published, this will include your full peer review and any attached files.

Reviewer #1: **Yes: **Francesco Venturelli

Reviewer #2: No

Reviewer #3: **Yes: **Glennis Margaret Andall-Brereton PhD, MPHCM,MPH

---

## [Author Response · Author response to Decision Letter 0]

21 Apr 2022

Thank you very much for considering our manuscript. We thank the reviewer and editorial staff for their valuable review. We carefully went through the constructive reviews provided by the peer reviewer and editor and have revised the manuscript accordingly. We have outlined our responses point by point to the reviewer’s comments. The line numbers presented here correspond to those in the final version (with track changes) of the manuscript. 

Reviewer 1: Thank you for the opportunity to revise the manuscript PONE-D-21-40413 “Severity of infection with the SARS-CoV-2 B.1.1.7 lineage among hospitalized COVID-19 patients in Belgium.” This is a very well written manuscript reporting results of a nationwide study on one of the more urgent topic in public health. The main value of the study is to establish a method that can be timely applied to assess the upcoming variant severity compared to previous circulating ones in hospitalized patients. In the current version of the manuscript, aims, methods, and results were clearly reported. Discussion and conclusions were based on presented results. Moreover, the major limitations and potential bias were deeply assessed and discussed, including the intrinsic limitations of hospital-based studies. I only have some minor comments, mainly related to clarifications on contextual factors.

Authors’ response: We thank the reviewer for the positive feedback and an accurate summary of the study.

Reviewer 1: The main comment is about indications to hospitalization in Belgium during the study period for COVID19 patients. As the authors pointed out, including only hospitalized patients is a limitation especially for external validity of study results. I think that a comment on potential differences in indications for hospitalization of COVID-19 patients between the two study periods and between younger and elderly patients could increase the understanding of the internal validity of study results.

Authors’ response: We thank the reviewer for this comment. Indeed, indications for hospitalization of COVID-19 patients may differ between settings (countries, regions, …) and may change over time. However, we consider the recommendations, guidelines or protocols in place (e.g., the ethical guidelines regarding the admission of COVID-19 patients ) to be comparable throughout the two time periods included in the current study (i.e., the second and third wave of the COVID-19 epidemic in Belgium). We believe that the indications for hospitalization mainly vary from hospital to hospital based on the availability of beds and staff to treat COVID-19 patients. Therefore, we have matched the two exposure groups based on the hospital and the mean ICU occupancy rate in order to account for differences in levels of care and admission criteria. Further, the decision-making process to admit COVID-19 patients may also be influenced by characteristics of the individual patient (e.g., age). Age differences between both exposure groups were taken into account by including age as a covariate in the multiple regression analysis and by performing a stratified analysis based on the age group. We have added the following on L484-497: “Further, although there were no apparent changes in national or regional policies, there may exist differences in indications for hospitalization of COVID-19 patients between the two time periods related to the number of available beds and medical personnel. However, we believe that matching the exposure groups based on the mean ICU occupancy rate (calculated as the number of COVID-19 patients occupying the recognized ICU beds within the hospital in which the patient was admitted and averaged over the patient’s hospital stay) controlled well for the risk of hospital or ICU admission related to organizational characteristics. In addition, matching on the hospital enables to account for between-hospital differences in admission criteria and levels of care. Moreover, the decision-making process to admit COVID-19 patients may also be influenced by individual patient characteristics such as age. Therefore, a major strength of the current study in general is the ability to control for an extensive list of potential confounders (i.e., patient characteristics and contextual factors that differ between the two time periods) given the detailed patient information that is collected within the CHS and the linkage to other data sources.”

Reviewer 1: Using the ICU bed occupancy rate as matching variable is a good strategy to reduce the impact of healthcare related factors on differences in patients’ outcomes. Notwithstanding, the absolute number of ICU beds even in the same hospital varied greatly across the pandemic periods, with an increase during each wave and generally inversely correlated with the quality of healthcare assistance provided (i.e. for healthcare services overload). Can you make a comment on the potential impact of this phenomenon on study results considering that it goes in the direction of higher risk for ICU admission among cases with B.1.1.7 variant, especially young?

Authors’ response: We thank the reviewer for this comment. We agree that the creation of additional ICU beds negatively impacts healthcare quality, as was demonstrated in a study using the Belgian Clinical Hospital Survey data from the first wave (Taccone, Van Goethem et al. 2020). Although additional beds were created during surge periods, qualified and trained staff could not be added on such quick notice and therefore using the recognized number of beds (instead of the sum of recognized plus newly created beds) makes sense as an indicator of healthcare quality. Thanks to the linkage with the exhaustive Surge Capacity Survey, we were able to calculate for each patient the mean ICU occupancy rate during his or her hospital stay, i.e., for the hospital in which the patient was hospitalized and during the patient’s hospital stay, we calculated the average number of COVID-19 patients occupying the hospital’s recognized ICU beds. A probably even better indicator for healthcare quality would be the nurse to patient ratio or the ICU physician to patient ratio but this information was not readily available in our linked health information system. Thus we believe that a balanced ICU occupancy rate of recognized beds (obtained through matching) between the exposed and unexposed controlled well for the risk of ICU admission. However, we have added the following to the Discussion section (L500-501): “As a limitation, we missed information on the staff to patient ratio and could not take into account the number of newly created ICU beds per hospital.” 

Reviewer 1: Despite very interesting, the chapter reporting the E-value and multi-bias E-value results is not straightforward. Could you rewrite it or expand the related paragraph in the discussion section explaining how these results relate with the robustness of the study results?

Authors’ response: We thank the reviewer for this comment. The interpretation of the E-value was elaborated on in the Methods section (L215-218): “The E-value is defined as the minimum strength of association, on the risk ratio scale, that an unmeasured confounder would need to have with both the treatment and the outcome to fully explain away a specific treatment-outcome association, conditional on the measured covariates.” as well as in the Results section (L356-371): “The E-value and multi-bias E-value were calculated to assess the influence of selection bias (e.g., based on the viral load) and/or unmeasured confounding (e.g., genetic profile of the patient) on the observed RR for each of the outcomes (S3 Table). The observed significant RR of 1.36 for ICU admission could be explained by an unmeasured confounder that was associated with both the exposure (SARS-CoV-2 variant) and ICU admission by a RR of 2.06-fold each, above and beyond the measured confounders, but weaker confounding could not do so; the confidence interval could be moved to include the null by an unmeasured confounder that was associated with both the exposure and ICU admission by a RR of 1.21-fold each, above and beyond the measured confounders, but weaker confounding could not do so. The same applies to selection on a variable with associations to both exposure and ICU transfer of at least 2.06 (1.21 for the 95% CI). A multi-bias E-value of 1.60 was obtained when considering both unmeasured confounding and selection bias simultaneously. This means that an unmeasured confounder with an association on the RR-scale of at least 1.60 to both exposure and outcome and selection on a variable with an association on the RR-scale of at least 1.60 to both exposure and outcome could explain the observed effect (above and beyond the variables that were controlled for in the model).” We have rewritten and added the following explanations on its interpretation in the Discussion section (L428-447) : Still, selection bias may arise when the samples from ICU patients are preferentially selected for WGS. Indeed, if a nonrandom selection of samples for WGS based on the severity of disease or ICU admission occurs, this may partially explain why we observed a higher standardized risk for ICU admission for patients with a confirmed B.1.1.7 infection compared to patients without available sequencing results that were considered to be infected with previously circulating strains. However, a sensitivity analysis considering only sequencing results obtained through baseline (unbiased) surveillance provided similar results. Another potential source of bias is the fact that only samples with a sufficiently high viral load (≥103-104 RNA copies/mL) can be sequenced due to technical limitations. This could bias our conclusions, as a higher viral load can be associated with severe disease [42]. However, the viral load also depends on the stage in which the patient is sampled (time of sampling) and the underlying conditions of the patients. Here, the robustness of our obtained causal inference estimates to potential uncontrolled confounding, such as the viral load, was assessed using the E-value [60]. If both the association between viral load and exposure (i.e. SARS-CoV-2 variant) and the association between viral load and ICU transfer, is at least 2.06 on the risk ratio scale (conditional on the other included covariates), this could completely nullify the observed causal estimate (RR = 1.36, 95% CI [1.03 – 1.68]) to be admitted in ICU. This relatively large E-value implies that considerable unmeasured or uncontrolled confounding would be needed to explain away our obtained effect estimate.

Reviewer 1: I would like to conclude by congratulating the study authors for the impressive work done.

Authors’ response: We thank the reviewer for taking the time to read our work and for these kind words.

Reviewer 2: It's an excellent paper that makes use of large-scale surveillance data to determine the severity of variants of concern for omicron sublineage B.1.1.7 in comparison to previously circulating strains. The findings are extremely informative, and the data was rigorously analysed. It's interesting that the authors made a concerted effort to provide "the most valid estimates" possible by adjusting for confounding variables using either matching or DAG, as used in this work. This article should be published to provide additional evidence regarding the severity of B.1.1.7 infection in a hospital-based setting.

Authors’ response: We thank the reviewer for taking the time to read our work, for providing an accurate summary, and for these kind words.

Reviewer 3: Overall the manuscript is very well written. However, there is need for some corrections and edits.

Authors’ response: We thank the reviewer for taking the time to review our manuscript and the valuable suggestions and corrections that were made. We have adapted all suggestions and corrections in the manuscript accordingly (see below).

RESULTS

277-279 - (Include Confidence Intervals (CI) in Table 1, where relevant)

Authors’ response: We thank the reviewer for this suggestion. We have opted to present continuous variables in Table 1 using a median accompanied by the interquartile range (IQR) as a measure of the spread of the data. Further, Table 1 was intended to provide the reader with a description of the characteristics of the study population and show if the two exposure groups are similar or different, and in what ways. There was no intention to perform statistical inference.

325-326 - There was no significant increased risk of in-hospital in neither of the age groups (<65 or > 65) - (review language)

Authors’ response: We thank the reviewer for the comment, it has been adjusted. L327-328: “There was no significant increased risk of in-hospital mortality in neither of the age groups (≤65 or >65 years)”

382-384 – When stratifying……,patients in general hospitals with B.1.1.7 infection were more frequently admitted in ICU (20.9 vs. 13.7%). - (include CI)

Authors’ response: We thank the reviewer for the comment, the confidence intervals for the proportions have been added. Also, we have now improved the readability of this sentence. L385-390: “When stratifying per hospital type, patients in general hospitals with viral sequence data were more frequently admitted into ICU as compared to patients without viral sequence data (20.9%; 95% CI [15.9% – 26.8%] and 13.7%, 95% CI [12.8% – 14.5%], respectively), whereas this difference was not observed among patients admitted to general hospitals with university characteristics or university hospitals.”

DISCUSSION

403 - However, restricting the analysis to hospitalized patients may include collider bias. - (include reference)

Authors’ response: We thank the reviewer for the comment. Two references have been added:

70. Munafò MR, Tilling K, Taylor AE, Evans DM, Davey Smith G. Collider scope: when selection bias can substantially influence observed associations. Int J Epidemiol. 2018;47: 226–235. doi:10.1093/ije/dyx206

71. Griffith GJ, Morris TT, Tudball MJ, Herbert A, Mancano G, Pike L, et al. Collider bias undermines our understanding of COVID-19 disease risk and severity. Nat Commun. 2020;11: 5749. doi:10.1038/s41467-020-19478-2

441 - ……..whereas severity seemed to be independent on the SARS- CoV-2 variant among the elderly- (review language)

Authors’ response: Thank you. It has been adjusted. L457-458: “…whereas severity seemed to be independent of the SARS-CoV-2 variant among the older age group (>65 years).”

481-484 - A major focus of the current genomic surveillance program is on detecting new emerging variants and flagging…………and geographic dynamics by monitoring returning travelers. - (Include reference)

Authors’ response: Thank you. A reference has been added.

75. RAG subgroup testing. Aanbevelingen voor de selectie van stalen voor de sequentiebepaling van het volledige genoom in het kader van surveillance – update. 2021. Available: https://covid-19.sciensano.be/sites/default/files/Covid19/20210315_Advice%20RAG_Selection%20for%20samples%20for%20sequencing%20-%20update_NL.pdf

CONCLUSIONS

497-498 - …….whereas severity was independent from the SARS- CoV-2 among 65-plus patients. (review language - needs clarifying for consistency- see 441-443)

Authors’ response: Thank you. It has been adjusted. L527-528: “…whereas severity was independent of the SARS-CoV-2 variant among the older age group (>65 year)”

500-501 - While at the moment of writing the delta-variant has completely dominated the B 1.1.7 variant, (Include ref.) this analysis……

Authors’ response: Thank you. A reference has been added.

76. National Reference Laboratory. Genomic surveillance of SARS-CoV-2 in Belgium. 2021 Dec. Available: https://assets.uzleuven.be/files/2021-12/genomic_surveillance_update_211214.pdf

502-503 - Real-time and unbiased analyses of emerging SARS-CoV-2 variants and their association with disease severity should be foreseen in the future. - (review language).

Authors’ response: Thank you. We have rephrased the sentence. L533-535: “Performing real-time and unbiased assessments of the severity related to emerging SARS-CoV-2 variants should be foreseen in the future.”

Other changes made to the manuscript:

After submitting this manuscript in December 2021, observations from other analyses using the socio-economic variables from the Statistics Belgium database pointed towards inconsistencies in the variable “household taxable income decile”. Therefore, we have decided to exclude this variable from the current analysis. We reran the analyses using the same fixed study population and analysis scripts as described in the Methods section, but using a multivariable model without including the covariate “household taxable income decile”. This resulted in slightly different effect estimates and confidence intervals in Table 2, Table 3, Supplementary Table 1, and Supplementary Table 2. Likewise, the numbers in the text related to the causal inference estimated and confidence intervals have been adapted (in track changes) accordingly. These minor numerical differences have no impact on the conclusions. Also, the analysis still accounts for socio-economic differences by the other remaining included covariates on the education level, the population density, and the median taxable income in the postcode of residence. We have adapted this in the Methods section (L194-196) and removed the “household taxable income decile” from Table 1 and Supplementary Table 4.

---

## [Decision Letter · Decision Letter 1]

16 May 2022

Severity of infection with the SARS-CoV-2 B.1.1.7 lineage among hospitalized COVID-19 patients in Belgium

PONE-D-21-40413R1

Dear Dr. Van Goethem,

We’re pleased to inform you that your manuscript has been judged scientifically suitable for publication and will be formally accepted for publication once it meets all outstanding technical requirements.

Kind regards,

Valérie Pittet, PhD

Academic Editor

PLOS ONE

Additional Editor Comments (optional):

Reviewers' comments:

Reviewer's Responses to Questions

**Comments to the Author**

1. If the authors have adequately addressed your comments raised in a previous round of review and you feel that this manuscript is now acceptable for publication, you may indicate that here to bypass the “Comments to the Author” section, enter your conflict of interest statement in the “Confidential to Editor” section, and submit your "Accept" recommendation.

Reviewer #1: All comments have been addressed

Reviewer #3: All comments have been addressed

2. Is the manuscript technically sound, and do the data support the conclusions?

Reviewer #1: Yes

Reviewer #3: Yes

3. Has the statistical analysis been performed appropriately and rigorously? 

Reviewer #1: Yes

Reviewer #3: Yes

4. Have the authors made all data underlying the findings in their manuscript fully available?

Reviewer #1: Yes

Reviewer #3: Yes

5. Is the manuscript presented in an intelligible fashion and written in standard English?

Reviewer #1: Yes

Reviewer #3: Yes

6. Review Comments to the Author

Reviewer #1: Dear Authors,

congratulations for your meaningful manuscript.

I think that in the current form it provides an added value to existing literature.

Best regards

Reviewer #3: (No Response)

7. PLOS authors have the option to publish the peer review history of their article (what does this mean?). If published, this will include your full peer review and any attached files.

Reviewer #1: **Yes: **Francesco Venturelli

Reviewer #3: **Yes: **Glennis Andall-Brereton

---

## [Editor Report · Acceptance letter]

23 May 2022

PONE-D-21-40413R1 

Severity of infection with the SARS-CoV-2 B.1.1.7 lineage among hospitalized COVID-19 patients in Belgium 

Dear Dr. Van Goethem:

I'm pleased to inform you that your manuscript has been deemed suitable for publication in PLOS ONE. Congratulations! Your manuscript is now with our production department. 

Kind regards, 

on behalf of

PD Dr. Valérie Pittet 

Academic Editor

PLOS ONE